# Impact of Pregestational Obesity on the Oral Health-Related Quality of Life in Brazilian Pregnant Women: A Cohort Study

**DOI:** 10.3390/ijerph21060740

**Published:** 2024-06-06

**Authors:** Ana Carolina da Silva Pinto, Gabriela de Figueiredo Meira, Francisco Carlos Groppo, Fernanda Ruffo Ortiz, Gerson Foratori, Eduardo Bernabé, Silvia Helena de Carvalho Sales-Peres

**Affiliations:** 1Department of Pediatric Dentistry, Orthodontics and Collective Health, Bauru School of Dentistry, University of São Paulo, São Paulo 17012-901, SP, Brazil; ana.pin@usp.br (A.C.d.S.P.); gabrielameira@usp.br (G.d.F.M.); gersonforatori.usp@gmail.com (G.F.J.); 2Department of Physiological Sciences, Faculty of Dentistry of Piracicaba, University of Campinas, Piracicaba 13414-903, SP, Brazil; fcgroppo@unicamp.br; 3School of Dentistry, ATTITUS Education, Passo Fundo 99070-220, RS, Brazil; fernandaruffoortiz@gmail.com; 4Department of Dental Public Health, King’s College London, London WC2R 2LS, UK; eduardo.bernabe@kcl.ac.uk

**Keywords:** quality of life, pregnancy women, Body Mass Index

## Abstract

The oral health-related quality of life of pregnant women and its effects on health conditions are important topics to be investigated in scientific research. The objective of this study was to evaluate the impact of pre-pregnancy obesity on oral health-related quality of life (OHRQoL) in pregnant women. A prospective cohort study was carried out with 93 pregnant women who were evaluated in the 2nd trimester of pregnancy (T1) and after delivery (T2). The following were analyzed: dental caries (DMFT), OHRQoL (OHIP-14), anthropometric data (BMI), socioeconomic, demographic, oral hygiene behavioral habits and the use of dental services. Unadjusted and adjusted Poisson regression analyses were performed to determine the impact of predictors on OHRQoL. The results of the adjusted analysis showed lower education relative risk (RR) (1.37; 95%CI 1.02–1.83; <0.00), low income (RR 2.19; 95%CI 1.63–2.93; <0.00) and higher BMI pre-pregnancy (RR 1.03; 95% CI 1.01–1.04; <0.00) were associated with worse OHRQoL in postpartum pregnant women. Flossing was a predictor of better OHRQoL at T2 (RR 0.73; 95%CI 0.57–0.93; <0.01). Higher BMI, low education, low income and inadequate oral hygiene habits were predictors of worse OHRQOL of pregnant women after the birth of the baby.

## 1. Introduction

Pregestational overweight and obesity are associated with adverse birth outcomes and have been reported as predictors for increased risk of developing gestational diabetes, pre-eclampsia, and increased chances of infant morbidity and mortality [1,2]. Furthermore, adiposity during pregnancy is associated with low birth weight and prematurity, conditions that increase the risk of enamel developmental defects [2].

In the long term, maternal obesity can affect offspring, increasing the chances of developing cardiovascular diseases, metabolic syndrome, diabetes, cancer, and psychiatric disorders [2,3].

Oral health is recognized as an essential component of quality of life and is becoming a relevant problem for public health. Hormonal changes during pregnancy and non-adherence to oral hygiene increase the incidence of oral diseases in pregnant women. Thus, poor oral health negatively influences OHRQoL, as it causes pain and suffering, functional changes, and aesthetic, nutritional and psychological problems [4,5].

During pregnancy, there may be an increased desire for sweet and high-fat foods, changes in oral conditions, such as increased salivary acidity and reduced saliva production, in addition to fear of dental treatment. These changes, associated with reduced brushing frequency and increased food consumption, can aggravate or predispose women to dental caries as well as gum and periodontal diseases. As a result, pregnant women end up being more susceptible and have a higher prevalence of developing dental caries [6,7,8]. In this sense, a systematic review carried out by Butera et al. (2021) highlights that during this period there is a change in the oral microbiota due to the greater availability of progesterone used in bacterial metabolism, with the proliferation of pathological bacteria such as *Fusobacterium nucleatum*, *Aggregatibacter actinomycetemcomitans*, *Prevotella intermedia*, *Porphyromonas gingivalis* and *Tannerella*; therefore, controlling the microbiota is of great importance, and health professionals can consider the use of probiotics in this population [9].

It is important to highlight that multidisciplinary and comprehensiveness have become the basis for health care for pregnant women in Brazil. In the Unified Health System (SUS), measures to qualify women’s health care have been implemented as part of public health policies to guarantee access and continuous health care in the promotion, prevention and dental treatment actions which are essential components of prenatal care [10].

The study conducted by Pacheco et al. (2020) on oral health-related quality of life with 1777 pregnant women treated in the Brazilian Public Health System warns that pregnant women who had higher levels of education had a better quality of life. However, the authors suggest that further research be carried out with clinical variables in this population [11].

In the literature, the relationship between pre-pregnancy overweight and obesity and general and oral health conditions after pregnancy is still controversial, although the relationship can be explained by several paths such as behavioral changes such as a greater intake of foods rich in sugar, as well as higher concentration of acidogenic bacteria, associated with decreased oral hygiene habits. It is worth adding, socioeconomic factors are associated with less knowledge about preventive methods and less access to oral health services, and finally, social support and social capital have been associated with better oral conditions in pregnant women [5,11,12,13]. Therefore, longitudinal studies that can generate knowledge and information regarding the effect of these factors on the quality of life of pregnant women are relevant. Therefore, the objective of this study was to evaluate the impact of pre-pregnancy obesity on the oral health-related quality of life (OHRQoL) in pregnant women.

## 2. Materials and Methods

### 2.1. Study Design

This work followed the guidelines Strengthening the Reporting of Observational Studies in Epidemiology (STROBE) for longitudinal studies [14].

This is a prospective cohort study in which data were collected in the preterm period ((T1) 2nd trimester) and post-term period (up to 2 months after the child’s birth (T2)) in the Basic Units of Health in Bauru, São Paulo, Brazil.

### 2.2. Ethical Aspects

According to the Declaration of Helsinki and Good Clinical Practice guidelines, this study was previously approved by the Ethics Committee on Human Research from the Bauru School of Dentistry—University of São Paulo CAAE 58339416.4.0000.5417. All participants provided written informed consent.

### 2.3. Sample and Sample Calculation

The sample was composed of pregnant women consecutively recruited from the Primary Health Care from Bauru, São Paulo, Brazil between October/2016 and June/2017 and follow-up occurred at least 3 months after delivery, until February/2018. The sample size calculation was performed considering the following parameters for the logistic regression model: 5% standard error, 95% confidence level, and seven independent variables (age, education, income, BMI, tooth brushing frequency, flossing frequency, dental visits and dental caries). Considering 50% the lowest expected proportion with an outcome, resulting in 140 pregnant women, statistical power was 80%, and 20% was added to compensate possible losses.

The inclusion criteria were pregnant women during the 2nd trimester of pregnancy (from the 12th gestational week), 18–40 years old, with regular prenatal care with obstetricians from Primary Health Care in Bauru, with adequate cognitive function, and without impairments that request absolute rest. Patients with neuromotor communication difficulties; who were diabetic and/or decompensated hypertensive; with malnutrition (BMI < 18 kg/m^2^); illicit drugs or alcohol users; smokers; those who presented with severe gestational problems requiring absolute rest; and users of medication that could interfere with the periodontal response (e.g., immunosuppressive, anticonvulsant or calcium channel-blocking drugs, such as cyclosporine, phenytoin or nifedipine, respectively) were excluded from the sample. 

A total of 166 pregnant women were initially recruited during the second trimester of pregnancy (T1). Of them, 3 pregnant women presented some impairment that required absolute rest, 2 were twin pregnancies and 1 was diagnosed with psychological pregnancy. At follow-up (T2), 67 participants did not attend, with the following reasons: no justification (n = 48), gave up participation in the research (n = 16); miscarriage (n = 2); and death of the baby without justification (n= 1) (Figure 1). Thus, the final sample was composed of 93 pregnant women.

### 2.4. Non-Clinical Data Collection

#### 2.4.1. Pregestational Obesity

Pregestational obesity (T1) was assessed according to the criteria and standards considered by the World Health Organization (WHO), in accordance with previous studies [10,15]. Body Mass Index (BMI) and weight (kg) was obtained from the patient’s medical record before pregnancy, and height was measured using a calibrated stadiometer (Wood 2.20; WCS Ind., Curitiba, Paraná, Brazil). An evaluator (ANP) was trained to measure height and body weight. When this data was absent from the medical record, the patients were weighed. To obtain the weight, an automatic scale was used (MIC model 300PP, Micheletti Ind Brazil maximum capacity 300 kg). Women with a BMI ≥ 25.00 kg/m^2^ were considered pre-pregnancy obese or obese when the collection was carried out after birth (T2).

#### 2.4.2. Socioeconomic Variables

The level of education was self-reported in the first consultation and recorded in the personal and general data sheet (T1) through the years of study. It was later dichotomized into high school (corresponding to 12 years of study in Brazil) and higher education (complete graduation). Monthly income was categorized in two levels: Low (≤3 Brazilian minimum wage) and High (>3 Brazilian minimum wage). The amount of BRL 937.00 (approximately USD 242.00) was considered the minimum wage specified by the Brazilian government in that year.

#### 2.4.3. Behavioral Habits Related to Oral Hygiene and Visits to the Dentist

The oral hygiene behaviors were as follows: daily use of dental floss, with the answer option “yes or no”, and the daily frequency of tooth brushing, with the answer options being 1 to 2 times and 3 times or more. The visit to the dentist was evaluated through the following question: frequency of visits to the dentist, with the answer options: never, caused by pain, and every 6 months.

#### 2.4.4. Oral Health-Related Quality of Life

The OHRQoL was considered the outcome of this study, and it was measured using a validated questionnaire and adapted for the Brazilian population [16]. The Oral Health Impact Profile (OHIP-14) was applied twice (T1 and T2). This questionnaire consists of 14 questions and 7 domains: functional limitation (did you have trouble pronouncing a word because of problems with your mouth, teeth or gums?/have you felt that the taste of food has become worse because of problems with your mouth, teeth or gums?); physical pain (have you felt pain in your mouth, teeth or gums?/have you felt uncomfortable eating any food because of problems in your mouth, teeth or gums?); psychological discomfort (were you worried because of problems with your mouth, teeth or gums?/have you felt stressed because of problems with your mouth, teeth or gums?); physical disability (has your diet been compromised due to problems with your mouth, teeth or gums?/have you had to stop eating because of problems with your mouth, teeth or gums?); psychological disability (have you found it difficult to relax because of problems with your mouth, teeth or gums?/did you feel a little embarrassed because of problems with your mouth, teeth or gums?); social incapacity (have you become angry with other people because of problems with your mouth, teeth or gums?/do you have difficulty carrying out your daily activities cause of problems with your mouth, teeth or gums?); and disability (have you felt that life in general has become worse because of problems with your mouth, teeth or gums?/have you been unable to carry out your daily activities because of problems with your mouth, teeth or gums?). Response codes were: 0 = never; 1 = rarely; 2 = sometimes; 3 = frequently; and 4 = always. The total score ranges from 0 to 28, where the highest values represent the worst OHRQoL [17]. 

### 2.5. Clinical Data Collection

The exams were performed by two examiners who were previously trained and calibrated, kappa intra-examiner (0.87) and inter-examiner (0.88), showing 10% of the sample for dental caries in T1 and T2, kappa intra-examiner (0.88) and inter-examiner (0.89). Pregnant women were examined at the health center itself on the same the day they had the schedule with the doctor confirmed. Gouzes were used for cleaning the surface of the teeth for better examination, and for the clinical exam, a plane mirror number 5 and WHO “ball point” probe were used, which were previously sterilized according to the standard biosecurity [18].

#### Dental Caries

Dental caries was evaluated using the Decayed, Missing and Filled permanent teeth index (DMFT) [12] in (T1) and (T2). All permanent teeth except the 3rd molars were evaluated.

The DMFT code has 10 records referring to the clinical condition observed in the teeth. In the present study, the clinical condition of each tooth except the 3rd molars was considered. The following codes were used: 0 = healthy tooth, 1 = decayed tooth, 2 = tooth restored but decayed, 3 = tooth restored and without decay, 4 = tooth lost due to decay, 5 = tooth lost for another reason, 6 = presence of sealant, 7 = bridge or crown support, 8 = unerupted tooth, T = trauma (fracture) and 9 = tooth excluded. The sum of teeth with codes 1 and 2 were considered decayed and teeth were considered lost by the sum of code 4 [19]. 

The prevalence of dental caries that was considered for the statistical analysis DMFT-D = 0 and DMFT-D > 0 and then decomposed for the evaluation of teeth lost due to caries and restored. 

### 2.6. Statistical Analysis

Data from the present study were tabulated and analyzed using the STATA 14.0 statistical program (Stata Corporation, College Station, TX, USA). For the descriptive analysis, the frequencies of the general characteristics of the sample and the total scores by domain of the OHRQoL questionnaire were estimated through means and standard deviation (SD). For data modeling, multivariate Poisson regression models were used to calculate the association between variables and outcomes (oral health-related quality of life). Variables that presented *p* ≤ 0.20 in the unadjusted analysis were included in the adjusted ones. Results are presented by relative risk (RR) and their respective 95% confidence intervals (95% CI) and statistical significance with *p* < 0.05.

## 3. Results

The characteristics and distribution of the sample are described in (Table 1). The sample consisted of 93 pregnant women. The mean BMI was 26.88 kg/m^2^ at baseline and 27.72 kg/m^2^ at follow-up. The average age of participants was 29.25 years (SD 5.22). Regarding income, 52.69% received more than minimum wage and 50.54% had completed higher education. Regarding behavioral habits, 80.65% of pregnant women performed tooth brushing 3 or more times a day and 19.35% performed it twice a day and 61.29% of participants used dental floss (T1). Of the evaluated pregnant women, 88 (94.62%) had a DMFT > 0. When the DMFT was individually analyzed, 4.4% were decayed, 77.17% were restored and 57.61% were lost. The average OHIP-14 score was 7.13 (SD 6.60) at baseline and 4.00 (SD 5.60) at follow-up.

The unadjusted and adjusted analysis between variables and overall OHIP-14 scores are shown in (Table 2). Pregnant women with dental caries, less education and lower monthly family income were at greater risk of having worse OHRQoL. Performing daily brushing 3 times or more had a protective effect on OHRQoL. The results of the adjusted analysis showed that lower education (RR 1.37; 95%CI 1.02–1.83), low income (RR 2.19; 95%CI 1.63–2.93) and higher pre-pregnancy BMI (RR 1.03; 95%CI 1.01–1.04) were statistically associated with greater impacts on the OHRQoL on pregnant women in the postpartum period. Higher daily frequency of dental flossing was a predictor of better OHRQoL at T2 (RR 0.73; 95%CI 0.57–0.93).

## 4. Discussion

Based on the results of the present study, it can be stated that women with pre-pregnancy obesity, with less education and low income, had greater oral impacts on OHRQoL.

It can also be observed that pregestational obesity is considered a factor that predisposes to greater weight gain during pregnancy, which can contribute to a risky pregnancy. The prevalence of obesity has increased not only in Brazil but worldwide [2,9], and it is important to highlight that this condition entails several comorbidities for individuals [10].

The results of our study showed that more than half of the participants were classified as obese. Data that agree with these results include the study by Seabra et al. [19] in which pregestational obesity was considered a risk factor for greater weight gain during pregnancy, and pregnant women with obesity had pre-eclampsia. Therefore, the authors suggest that nutritional educational measures should be a priority in prenatal care.

In the present study, we found a significant relationship between a DMFT > 0 and OHRQoL, generating a negative effect for a worse quality of life in pregnant women. This fact can be explained by the higher prevalence of restored teeth when analyzing the DMFT components individually. However, Rosell et al. [20] found that pregnant women with high DMFT scores had high OHRQoL. Psychological discomfort and physical pain were the most affected dimensions in the OHIP-14, which is to be expected since dental caries cause episodes of pain, which generates temporary or permanent inability to eat, communicate and carry out daily activities. A longitudinal study showed that high maternal BMI is associated with functional limitation, physical disability, and handicap during pregnancy [21]. 

Thus, the differences between the studies can be explained by the higher prevalence of active dental caries when compared to our findings, in which the restored component was greater.

In this study, there was a decrease in the frequency of brushing and oral hygiene care after the baby’s birth. This fact leads to the hypothesis that after birth, mothers stop taking care of themselves to offer greater care and attention to their newborn children. Thus, the results of the systematic review carried out by Xião et al. [22] reinforce the idea that those preventive measures should be encouraged during pregnancy, as women are more open to receiving information during this period. 

Also, in this study we found that lower income and low education negatively influenced the post-pregnancy OHRQoL of participants. These data agree with a recent systematic review with meta-analysis performed by Foratori-junior et al. [23] who found that the negative impacts on OHRQoL have been associated with a lower socioeconomic level and less education [18]. These findings can be explained by the fact that people with lower socioeconomic status are more susceptible to risk factors for oral diseases; in addition, low education has been associated with less knowledge about preventive measures and healthy habits.

It is also worth noting that the literature has shown that low income is linked to nutritional deficiencies such as being overweight or underweight, higher prevalence of chronic diseases, greater use of over-the-counter medications and risky behaviors such as tobacco use and alcohol consumption among pregnant women, which characterizes the vulnerability of these women with unfavorable socioeconomic conditions [10,19].

This study had some limitations. The presence of dental plaque was not measured using validated indicators, since dental caries is an oral disease induced by biofilm, despite this being an oral condition that is multifactorial and complex, which leads to pain and functional limitation. However, the hygienic status of the patients was measured through the frequency of tooth brushing and flossing. We can cite the measurements related to the weight of patients present in the medical records, which are subject to measurement error, so we cannot be sure that all women had that initial weight, but we believe that this does not have a major impact on our results since this data is collected by health professionals qualified to provide care in the Brazilian Public Health System.

Another limitation refers to the postpartum evaluation, which should have been carried out after at least 6 months, as breastfeeding has a hormonal influence on the body. Future studies with a larger sample size should be performed to assess the impact of dental caries and pregestational obesity on OHRQoL during pregnancy.

The lack of association between dental caries and pregnant women seems difficult to explain.

One hypothesis is that the initiation and progression of the caries lesion is very slow, and the destruction caused by caries in the initial stage can be reversible [21]. Untreated dental caries was not associated with preterm birth or pre-eclampsia but with the risk of delivering large-for-gestational-age infants. This fact may be attributed to the various characteristics of mothers who develop dental caries that are not treated [7]. Further studies are needed to evaluate the association between dental caries and adverse pregnancy outcomes in pregnant women and their quality of life. Maternal obesity and dental caries impact negatively on quality of life, increasing the risk factors for systemic and oral health, in prenatal and postnatal care.

Therefore, future longitudinal research and randomized clinical trials should be carried out with a focus on promoting oral health and reducing the incidence of dental caries in this population through the evaluation of hydroxyapatite biomimetics and the use of probiotics that ensure balance of the oral microbiota.

## 5. Conclusions

Dental caries affects the oral health-related quality of life of pregnant women with pregestational excess weight. After the birth of the child, there is a worsening in the quality of life related to the oral health of these women. In addition, higher BMI, low education, low income and inadequate oral hygiene habits were predictors of worse quality of life for pregnant women after the birth of the child. Therefore, actions that reduce the risk of tooth decay should be encouraged, such as oral hygiene guidelines during prenatal care, consumption of foods with low sucrose content and regular visits to the dentist.

## Figures and Tables

**Figure 1 ijerph-21-00740-f001:**
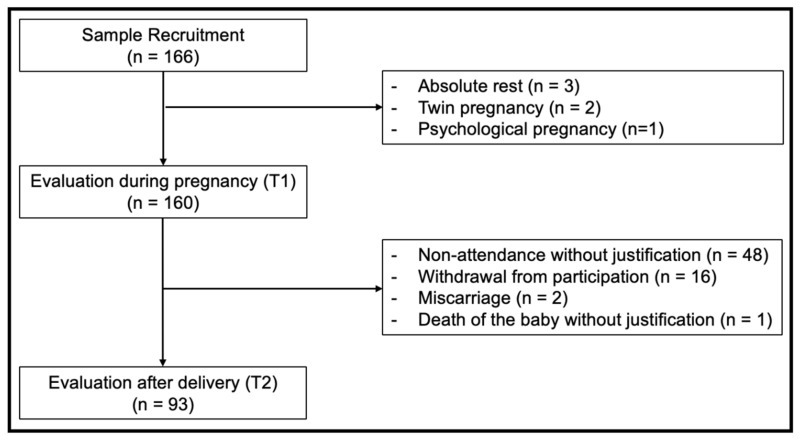
Flowchart for composition of the sample.

**Table 1 ijerph-21-00740-t001:** Descriptive analysis of the sample at baseline (T1) and follow-up (T2).

Variables	T1 n (%)	T2 n (%)
Age ^+^	29.25 (5.22)	
Education		
Higher	47 (50.54)	
High school	46 (49.46)	
Monthly income		
High	49 (52.69)	
Low	44 (47.31)	
BMI ^++^	26.88 (6.17) [17.58–56.27] ^++^	27.72 (5.87) [18.16–54.82] ^++^
No. of tooth brushing		
2 times per day	18 (19.35)	44 (47.31)
3 times or more	75 (80.65)	49 (52.69)
Flossing		
No	36 (38.71)	63 (67.74)
Yes	57 (61.29)	30 (32.26)
Visit to the dentist		
No	55 (59.14)	
Yes	38 (40.86)	
Dental caries	7.87 (5.14)	9.18 (5.34)
DMFT-D = 0	5 (5.38)	5 (5.38)
DMFT-D >0	88 (94.62)	88 (94.62)
OHIP-14	7.13 (6.60)	4 (5.60)

BMI: Body Mass Index; DMFT: decayed, missing and filled teeth; OHIP: Oral Health Impact Profile; T1: baseline; T2: follow-up. + Age: Mean (standard deviation), ++ BMI: minimum–maximum.

**Table 2 ijerph-21-00740-t002:** Unadjusted and adjusted Poisson regression analysis between factors associated with the overall OHIP-14 score.

Variables	Oral Health-Related Quality of Life (OHIP-14)			
	Unadjusted Analysis		Adjusted Analysis	
	RR (CI 95%)	*p* Value	RR (CI 95%)	*p* Value
Age	1.00 (0.98–1.02)	0.58		
Education				
Higher	1		1	
High school	2.59 (2.07–3.25)	<0.00 *	1.37 (1.02–1.83)	0.04 *
Monthly income				
High	1		1	
Low	2.96 (2.36–3.72)	<0.00 *	2.19 (1.63–2.93)	<0.00 *
BMI	1.05 (1.03–1.06)	<0.00 *	1.03 (1.01–1.04)	<0.00 *
No. of tooth brushing				
2 times per day	1			
3 times or more	0.88 (0.72–1.08)	0.25		
Flossing				
No	1		1	
Yes	0.66 (0.52–0.84)	<0.00 *	0.73 (0.57–0.93)	0.01 *
Visit to the dentist				
No	1			
Yes	1.08 (0.88–1.33)	0.43		
Dental caries				
DMFT = 0	1		1	
DMFT > 0	0.71 (0.48–1.05)	0.09	0.67 (0.45–0.98)	0.04 *

BMI: Body Mass Index; DMFT: decayed, missing and filled teeth; OHIP-14: Oral Health Impact Profile; RR: relative risk, * *p* < 0.05.

## Data Availability

The data presented in this study are available from the Southern Community Cohort Study upon request. The data are not publicly available to protect participant privacy.

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
