# Peer review of "Impact of Pregestational Obesity on the Oral Health-Related Quality of Life in Brazilian Pregnant Women: A Cohort Study"

_ijerph, 2024, doi:10.3390/ijerph21060740_

Round 1
Reviewer 1 Report (Previous Reviewer 1)
Comments and Suggestions for Authors
The authors have addressed all of my previous comments in the revised version of the manuscript. I recommend its acceptance for publication in the International Journal of Environmental Research and Public Health.
Comments on the Quality of English LanguageMinor editing of English language required.
Author Response
We are grateful for the considerations made to improve the manuscript and we are immensely grateful for publishing and being able to contribute to studies in a journal with great international impact in the area of public health.
Reviewer 2 Report (Previous Reviewer 2)
Comments and Suggestions for Authors
The authors are submitting their study titled "Impact of pregestational obesity on the oral health-related quality of life in Brazilian pregnant women: a cohort study" for the third time. They have implemented several corrections in contrast to the initial version of the study. Nonetheless, a principal and substantial flaw persists due to the absence of data on the dental status of the patients before the onset of pregnancy. Additionally, in the current iteration, the authors introduce data regarding the DMFT (Decayed, Missing, Filled Teeth) index being 7.87. This detail allows for the inference that the level of oral cavity sanitation in patients prior to pregnancy was notably low. Consequently, the assertion regarding the impact of pregnancy on Oral Health-Related Quality of Life (OHRQoL) becomes even more dubious. Without the presence of baseline data (dental status before pregnancy), the study essentially provides merely descriptive data pertaining to a specific time frame for a certain patient group.
Author Response
Dear Reviewer,
It is with great satisfaction that we obtained approval for publication by 3 of the 4 reviewers, after the adjustments and clarifications made.
Considering your questions, we will raise some clarifications regarding the points mentioned, since as a solid team in this line of research, we do not want any doubts to arise regarding the data found. It is worth highlighting that we have 13 publications in the area involving monitoring pregnant women, during pregnancy and after birth.
We are from a renowned teaching, research and extension institution in the world, in order to achieve excellence, all precautions to conduct robust research were certainly based on scientific evidence and with the possibility of opening new questions for future studies. Our work is based on Public Health and highlights the importance of findings for decision-making at the level of health professionals, researchers and health system managers. As we know, health is not an exact science, subject to limitations, doubts and teachings. When we involve human beings and their limitations, such as the impossibility of detecting the exact period before pregnancy, this may be partially a limitation, however, it directs us to the best care to be offered to these women during this period.
Initially, I would like to remember the limitations that have already been mentioned in other reviews of this article, such as the measurement of dental plaque. It is worth remembering that although this variable was not collected, other indicators that lead to tooth decay were included, such as the frequency of tooth brushing.
The oral health status of the participants was mentioned in the study through the following excerpt that can be found in the manuscript:
“Dental caries was evaluated using the Decayed, Missing and Filled permanent teeth index (DMFT) [12] in (T1) and (T2). All permanent teeth except 3rd molars were evaluated.
The DMFT code has 10 records referring to the clinical condition observed in the teeth. In the present study, the clinical condition of each tooth except the 3 molars was considered. The code 0 healthy tooth, 1 decayed tooth, 2 tooth restored but decayed, 3 tooth restored and without decay, 4 tooth lost due to decay, 5 tooth lost for another reason, 6 presence of sealant, 7 bridge or crown support, 8 unerupted tooth, T trauma (Fracture) and 9 tooth excluded. The sum of teeth with codes 1 and 2 were considered decayed and teeth lost by the sum of code 4 [19].
The prevalence of dental caries that was considered for the statistical analysis DMFT-D = 0 and DMFT-D > 0 and then decomposed for the evaluation of teeth lost due to caries and restored”
We would like to mention again some studies carried out by the research group that mention the importance of studying these factors in the quality of life related to the oral health of pregnant women.
FORATORI JUNIOR, G. A. ; Ventura, T.M.O. ; GRIZZO, L. T. ; JESUIN, B. G. ; CASTILHO, A. V. S. S. ; BUZALAF, Marilia Afonso Rabelo ; SALES PERES, SILVIA HELENA DE CARVALHO . Is There a Difference in the Proteomic Profile of Stimulated and Unstimulated Saliva Samples from Pregnant Women with/without Obesity and Periodontitis?. Cells, v. 12, p. 1389, 2023.
FORATORI-JUNIOR, GERSON APARECIDO ; LE GUENNEC, ADRIEN ; FIDALGO, TATIANA KELLY DA SILVA ; JARVIS, JAMES ; MOSQUIM, VICTOR ; BUZALAF, Marília Afonso Rabelo ; CARPENTER, GUY HOWARD ; SALES-PERES, SILVIA HELENA DE CARVALHO . Comparison of the Metabolic Profile between Unstimulated and Stimulated Saliva Samples from Pregnant Women with/without Obesity and Periodontitis. Journal Of Personalized Medicine, v. 13, p. 1123, 2023.
DAMANTE, CARLA ANDREOTTI ; FORATORI JUNIOR, G. A. ; CUNHA, P. A. ; NEGRATO, C. A. ; SALES PERES, SHC ; ZANGRANDO, M. S. R. ; SANTANA, A. C. P. . Association among gestational diabetes mellitus, periodontitis and prematurity: a cross-sectional study. ARCHIVES OF ENDOCRINOLOGY AND METABOLISM, v. 66, p. 58-67, 2022.
FORATORI JUNIOR, G. A. ; PEREIRA, P. R. ; GASPAROTO, I. A. ; SALES PERES, SHC ; SOUZA, J. M. S. ; KHAN, S. . Is overweight associated with periodontitis in pregnant women? Systematic review and meta-analysis. Japanese Dental Science Review, v. 58, p. 41-51, 2022.
FORATORI JUNIOR, G. A. ; Ventura, T.M.O. ; Thomassian L. T..G ; CARPENTER, G.H. ; BUZALAF, M ; SALES PERES, SILVIA HELENA DE CARVALHO . Label-free quantitative proteomic analysis reveals inflammatory pattern associated with obesity and periodontitis in pregnant women. METABOLITES, v. 12, p. 1091, 2022.
FORATORI JUNIOR, G. A. ; GUENNEC ; FIDALGO, T. K. S. ; CLEAVER, L. ; BUZALAF, M.A.R. ; CARPENTER, G. ; SALES-PERES, S. H. DE C. . Metabolomic profiles associated with obesity and periodontitis during pregnancy: cross- sectional study with Proton Nuclear Magnetic Resonance (1H-NMR)-based analysis. METABOLITES, v. 12, p. 1029, 2022.
FORATORI-JUNIOR, GERSON APARECIDO ; MISSIO, ALANA LUIZA TRENHAGO ; ORENHA, ELIEL SOARES ; de Carvalho Sales-Peres, Silvia Helena . Systemic Condition, Periodontal Status, and Quality of Life in Obese Women During Pregnancy and After Delivery. INTERNATIONAL DENTAL JOURNAL, v. 71, p. 1-9, 2021.
FORATORI-JUNIOR, GERSON APARECIDO ; MOSQUIM, VICTOR ; RABELO BUZALAF, MARÍLIA AFONSO ; HELENA DE CARVALHO SALES-PERES, SILVIA . Salivary cytokines levels, maternal periodontitis and infants' weight at birth: a cohort study in pregnant women with obesity. PLACENTA, v. 115, p. 151-157, 2021.
CARACHO, R. A. ; FORATORI JUNIOR, G. A. ; FUSCO, N. S. ; JESUINO, B. G. ; MISSIO, A. L. T. ; DE CARVALHO SALES-PERES, SH . Systemic condition and oral health-related quality of life in pregnant women with and without overweight assisted by public healthcare system. INTERNATIONAL DENTAL JOURNAL, v. 70, p. 287-295, 2020.
FORATORI JUNIOR, G. A. ; JESUINO, BRUNO GUALTIERI ; CARACHO, R. A. ; Orenha, ES ; Groppo FC ; SALES-PERES, SHC . Association between excessive maternal weight, periodontitis during the third trimester of pregnancy, and infants' health at birth. Journal of Applied Oral Science, v. 28, p. e20190351, 2020
FORATORI-JUNIOR, GERSON APARECIDO ; DA SILVA, BRUNA MACHADO ; DA SILVA PINTO, ANA CAROLINA ; HONÓRIO, HEITOR MARQUE ; GROPPO, FRANCISCO CARLOS ; de Carvalho Sales-Peres, Silvia Helena . Systemic and periodontal conditions of overweight/obese patients during pregnancy and after delivery: a prospective cohort. Clinical Oral Investigations, v. 24, p. 157-165, 2020.
JESUINO, BRUNO GUALTIERI; FORATORI'JUNIOR, GERSON APARECIDO ; MISSIO, ALANA LUIZA TRENHAGO ; MASCOLI, LEONARDO SILVA ; SALES'PERES, SILVIA HELENA DE CARVALHO . Periodontal status of women with excessive gestational weight gain and the association with their newborns? health. INTERNATIONAL DENTAL JOURNAL, v. 70, p. 396-404, 2020.
FUSCO, NATHALIA DOS SANTOS ; FORATORI'JUNIOR, GERSON APARECIDO ; MISSIO, ALANA LUIZA TRENHAGO ; JESUINO, BRUNO GUALTIERI ; SALES'PERES, SILVIA HELENA DE CARVALHO . Systemic and oral conditions of pregnant women with excessive weight assisted in a private health system. INTERNATIONAL DENTAL JOURNAL, v. 69, p. idj12507, 2019.
I would like to include in this answer that the study follows the theoretical model for quality of life related to oral health proposed by Wilson and Cleary, 1995, which seeks to include explanatory variables for changes in quality of life related to health, where we can find oral diseases, the biological conditions of the individual, the behaviors to improve oral health, so through this study we tried to relate the variables of the model with the variables collected in the project from which this manuscript was the result.
Wilson IB, Cleary PD. Linking clinical variables with health-related quality of life. A conceptual model of patient outcomes. JAMA. 1995;273(1):59-65.
A recent publication Wilson A, Bridgman H, Bettiol S, Crocombe L, Hoang H. Bridging the evidence-to-practice gap: exploring dental professionals' perspectives on managing oral health during pregnancy in Tasmania, Australia. Aust Dent J. Published online May 15, 2024.
It reinforces the importance of our study, since it is not possible to predict the moment when a woman will become pregnant, therefore, studies on the prediction of factors related to pre-gestational and gestational changes should be encouraged at all scientific levels.
Reviewer 3 Report (Previous Reviewer 3)
Comments and Suggestions for Authors
Considering the previous revisions, the reviewer considers that the authors address the comments in the persent revised version.
Comments on the Quality of English LanguageMinor editing is advised.
Author Response
The authors are grateful for the considerations made by the reviewer to improve the manuscript
Reviewer 4 Report (New Reviewer)
Comments and Suggestions for Authors
Manuscript of medium interest for the dental sector, arginine was already studied in similar studies.
Abstract: Add statistical results
Few keywords: add specific ones recorded on MeSH.
Introductions: add all the causes of caries risk and how the oral microbiota changes during pregnancy (Butera et al).
Materials and methods; insert the new Consort Flow Chart
Results, very confusing, reorganize the tables to make them more usable for the reader and highlight the statistically significant data.
Discussion; add as future objectives the evaluation of biomimetic hydroxyapatite to reduce the incidence of caries risk. It's the future! and all natural systems that do not affect the dysbiosis of the oral cavity, such as probiotics, paraprobiotics, post biotics and ozonated gels (Scribante et al) cite his works.
Conclusions: add proactive action to reduce caries risk
Bibliography: add required references
Round 2
Reviewer 4 Report (New Reviewer)
Comments and Suggestions for Authors
The manuscript is suitable for publication
This manuscript is a resubmission of an earlier submission. The following is a list of the peer review reports and author responses from that submission.
Round 1
Reviewer 1 Report
Comments and Suggestions for Authors
This is a re-submission of the manuscript (ID: ijerph-2812033) that I have reviewed earlier, originally entitled “Impact of dental caries pregestational and obesity on the oral health-related quality of life on pregnant women: a cohort study”. This revised version of the manuscript has been somewhat improved. However, the following comments were not properly addressed in the revision:
- Please reword the term “obesity pregestational” to “pregestational obesity” or “prepregnancy obesity” in the title.
- Please reword the keyword “Mass body index” to “Body mass index”.
- I couldn’t find the original study questionnaires (as supplementary files).
- Did authors use radiography to assess and diagnose dental caries?
Comments on the Quality of English Language
Moderate editing of English language required
Author Response
-The article title has been corrected: Impact of pregestational obesity on the oral health-related quality of life in Brazilian pregnant women: a cohort study
Please reword the keyword “Mass body index” to “Body mass index”.
-Keyword corrected: Body mass index
- I couldn’t find the original study questionnaires (as supplementary files).
-The original questionnaires were included as a supplementary file
- Did authors use radiography to assess and diagnose dental caries?
- In this study, no x-rays were taken, as we followed the diagnostic criteria in accordance with the WHO recommendations for epidemiological surveys in oral health, which only the visual tactile clinical examination of the tooth surface is the diagnostic gold standard for this oral condition.
Reviewer 2 Report
Comments and Suggestions for Authors
I have thoroughly reviewed the resubmitted manuscript provided by the authors. Unfortunately, despite the clarification and specification added to this version, I regret to inform you that I still recommend rejecting the manuscript for publication. My assessment is grounded in several key concerns that were not adequately addressed by the authors.
“…..substantial shortcoming of this study lies in its failure to provide information about the dental status of the participants before pregnancy. Understanding the baseline oral health conditions of the subjects is paramount for establishing a clear correlation between pregnancy and its influence on oral health-related quality. Without this essential baseline data, it is challenging to discern the true relationship between pregnancy and oral health.
Furthermore, the manuscript lacks crucial details regarding the pregestational preparation of the respondents. This omission is particularly concerning, as the preparation for pregnancy plays a pivotal role in shaping the course of gestation and its effects on various bodily systems. A comprehensive examination of pregestational conditions and preparations is necessary to provide a comprehensive understanding of the research context….”
Overall, the primary flaw of the study is the lack of an assessment of the oral cavity's condition before the onset of pregnancy. The new hypothesis proposed by the authors ("Impact of pregestational obesity on the oral health-related quality of life in Brazilian pregnant women: a cohort study") also cannot be evaluated due to the absence of physiological status (body mass index) before pregnancy.
Furthermore, one of the leading factors in the development of caries and periodontal diseases is the hygienic condition of the oral cavity (presence of dental plaque, which is subject to index assessment). The inclusion of this factor was suggested to the authors in the first round of review, however, it was ignored.
Author Response
“…..substantial shortcoming of this study lies in its failure to provide information about the dental status of the participants before pregnancy. Understanding the baseline oral health conditions of the subjects is paramount for establishing a clear correlation between pregnancy and its influence on oral health-related quality. Without this essential baseline data, it is challenging to discern the true relationship between pregnancy and oral health.
Dear reviewer, this article is part of a large project which generated two master's theses, the first entitled: Assessment of periodontal condition and its impact on quality of life in patients with pre-pregnancy excess weight before and after childbirth, which served as the basis for publishing the article entitled Salivary cytokines levels, maternal periodontitis and infants’ weight at birth: A cohort study in pregnant women with obesity published in the Placenta, in 2021. doi: 10.1016/j.placenta.2021.09.018. In the article, other oral conditions were investigated during pregnancy and after birth, such as gingival bleeding, periodontal disease, salivary cytokines (IL-1β, TNF-α and leptin). The second dissertation entitled: Dental caries and quality of life in pregnant women, with and without pre-pregnancy excess weight: prospective cohort, on which this article is based on Impact of pregestational obesity on the oral health-related quality of life in Brazilian pregnant women: a cohort study. Therefore, other oral conditions were not included in the already published article. However, as a researcher, the relationship between pre-pregnancy and postpartum obesity, tooth decay and quality of life caught my attention. So, if it is not clear in the writing, we will correct it.
Dental examinations were carried out on patients who are part of the women's clinic in Basic Health Units (BHU) in the Bauru, São Paulo, Brazil at a time when women were seeking medical care in these places, thus, before pregnancy, tooth decay was measured by the DMFT and ICDAS (T1) indicators. At that time, the women were weighed and measured to assess their BMI, if these measurements were not in the medical record. When the weight was present in the patient's medical record, the evaluators only measured the height for BMI. Women attending in the BHU, have a medical record with clinical and exam data, however our interest was in anthropometric measurements.
In the article published, women who became pregnant were again invited to participate in the study and new oral conditions were evaluated. In the present study we consider the variables before pregnancy and after the birth of the child (T2).
Furthermore, the manuscript lacks crucial details regarding the pregestational preparation of the respondents. This omission is particularly concerning, as the preparation for pregnancy plays a pivotal role in shaping the course of gestation and its effects on various bodily systems. A comprehensive examination of pregestational conditions and preparations is necessary to provide a comprehensive understanding of the research context….”
The recruitment of participants took place during visits that were made to health centers in the Bauru-SP. Pregnant women were invited to participate in the research and if they accepted, the first consultation would take place at that time. At the time of the research carried out in 2017, we included variables that could meet all dimensions involving the health and disease process, such as sociodemographic (age), economic (education, monthly income), behavioral (toothbrushing), use of dental services), anthropometric measurement (BMI), Presence of tooth decay and quality of life related to oral health. We agree that other parameters are also important and will be included in new projects developed by the research group.
Overall, the primary flaw of the study is the lack of an assessment of the oral cavity's condition before the onset of pregnancy. The new hypothesis proposed by the authors ("Impact of pregestational obesity on the oral health-related quality of life in Brazilian pregnant women: a cohort study") also cannot be evaluated due to the absence of physiological status (body mass index) before pregnancy.
Item 2.4.1 was rewritten to better understand the physiological state at baseline and follow-up, as shown in the text below:
Pregestational obesity (T1) was assessed according to the criteria and standards considered by the World Health Organization (WHO), in accordance with previous studies [9,14], using Body Mass Index (BMI) and Weight (kg) was obtained from the patient's medical record. before pregnancy and height was measured using a calibrated stadiometer (Wood 2.20; WCS Ind., Curitiba, Paraná, Brazil). An evaluator (ANP) was trained to measure height and body weight. When this data was absent from the medical record, the patients were weighed. To obtain the weight, an automatic scale was used (MIC model 300PP, Micheletti Ind., maximum capacity 300kg). Women with a BMI ≥ 25.00 kg/m2 were considered prepregnancy obese or obese when the collection was carried out after birth (T2).
Furthermore, one of the leading factors in the development of caries and periodontal diseases is the hygienic condition of the oral cavity (presence of dental plaque, which is subject to index assessment). The inclusion of this factor was suggested to the authors in the first round of review, however, it was ignored.
We agree with the suggestion, the presence of dental plaque was not collected and was included as a limitation of the study, as shown in the text below:
This study had some limitations. The presence of dental plaque was not measured using validated indicators, since dental caries is an oral disease induced by biofilm, despite this being an oral condition that is multifactorial and complex, which leads to pain and functional limitation. However, the hygienic status of the patients was measured through the frequency of tooth brushing and flossing. We can cite the measurements related to the weight of patients present in the medical records, which are subject to measurement error, so we cannot be sure that all women had that initial weight, but we believe that this does not have a major impact on our results. since this data is collected by health professionals qualified to provide care in the Brazilian Public Health System.
Reviewer 3 Report
Comments and Suggestions for Authors
The reviewer considers that the following should be addresses.
1. Why the relation between prepregnancy overweight and obesity and 60 general and oral health conditions after pregnancy can be explained by several paths (line62)? Detailing information would improve that statement.
2. Please detail further about the impact of study´s results on clinical decision and how it adds to the current practice in study population.
Comments on the Quality of English LanguageMinor review is required
Author Response
- Why the relation between prepregnancy overweight and obesity and 60 general and oral health conditions after pregnancy can be explained by several paths (line62)? Detailing information would improve that statement.
A correction was made to the article, as shown in the text below:
In the literature, the relationship between pre-pregnancy overweight and obesity and general and oral health conditions after pregnancy is still controversial, although the relationship can be explained by several paths such as behavioral changes such as greater intake of foods rich in sugar, as well as higher concentration of acidogenic bacteria, associated with decreased oral hygiene habits. It is worth adding, socioeconomic factors are associated with less knowledge about preventive methods and less access to oral health services and finally, social support and social capital have been associated with better oral conditions in pregnant women [5,10-12]. Therefore, longitudinal studies that can generate knowledge and information regarding the effect of these factors on the quality of life of pregnant women are relevant. Therefore, the objective of this study was to evaluate the impact of pre-pregnancy obesity on oral health-related quality of life (OHRQoL) in pregnant women.
- Please detail further about the impact of study´s results on clinical decision and how it adds to the current practice in study population.
The results of the present study to lead recommended dental practice focused on the expanded clinic, reinforcing that dental professional need greater knowledge about systemic changes such as pre-gestational obesity, so that the treatment plan offers a better quality of life to the patient. Therefore, these efforts must be used in a united manner between obstetricians, doctors and dental surgeons.